# Exploring the Role of GMMA Components in the Immunogenicity of a 4-Valent Vaccine against *Shigella*

**DOI:** 10.3390/ijms24032742

**Published:** 2023-02-01

**Authors:** Francesca Mancini, Renzo Alfini, Valentina Caradonna, Valentina Monaci, Martina Carducci, Gianmarco Gasperini, Diego Piccioli, Massimiliano Biagini, Carlo Giannelli, Omar Rossi, Mariagrazia Pizza, Francesca Micoli

**Affiliations:** 1GSK Vaccines Institute for Global Health (GVGH), via Fiorentina 1, 53100 Siena, Italy; 2GSK, via Fiorentina 1, 53100 Siena, Italy

**Keywords:** generalized modules for membrane antigens (GMMA), *Shigella*, vaccine, OMV, TLR, proteins

## Abstract

Shigellosis is the leading cause of diarrheal disease, especially in children of low- and middle-income countries, and is often associated with anti-microbial resistance. Currently, there are no licensed vaccines widely available against *Shigella*, but several candidates based on the O-antigen (OAg) portion of lipopolysaccharides are in development. We have proposed Generalized Modules for Membrane Antigens (GMMA) as an innovative delivery system for OAg, and a quadrivalent vaccine candidate containing GMMA from *S. sonnei* and three prevalent *S. flexneri* serotypes (1b, 2a and 3a) is moving to a phase II clinical trial, with the aim to elicit broad protection against *Shigella*. GMMA are able to induce anti-OAg-specific functional IgG responses in animal models and healthy adults. We have previously demonstrated that antibodies against protein antigens are also generated upon immunization with *S. sonnei* GMMA. In this work, we show that a quadrivalent *Shigella* GMMA-based vaccine is able to promote a humoral response against OAg and proteins of all GMMA types contained in the investigational vaccine. Proteins contained in GMMA provide T cell help as GMMA elicit a stronger anti-OAg IgG response in wild type than in T cell-deficient mice. Additionally, we observed that only the trigger of Toll-like Receptor (TLR) 4 and not of TLR2 contributed to GMMA immunogenicity. In conclusion, when tested in mice, GMMA of a quadrivalent *Shigella* vaccine candidate combine both adjuvant and carrier activities which allow an increase in the low immunogenic properties of carbohydrate antigens.

## 1. Introduction

*Shigella* is a major cause of mortality and morbidity, especially in children under 5 years of age living in low-middle income countries. No vaccine is yet available globally, and the increased multidrug resistance [1] makes the introduction of a vaccine against *Shigella* an even higher global health priority [2]. Depending on the composition of the OAg, the genus *Shigella* is composed of 50 different serotypes divided into four serogroups: *S. sonnei* (one serotype), *S. flexneri* (15 serotypes), *S. dysenteriae* (15 serotypes) and *S. boydii* (19 serotypes) [3]. Not all serotypes cause severe illness. *S. sonnei* and *S. flexneri* are the biggest cause of diarrheal disease and the currently most widespread species [4].

The OAg component of the lipopolysaccharide (LPS) is involved in many interactions between the pathogen and host and has been recognized as a key protective antigen [5]. *Shigella* OAg-based vaccines are currently in clinical development and have demonstrated their immunogenicity [4,6,7]. Additionally, other virulence factors have been proposed as cross-protective antigens for vaccine development [8], such as proteins of the type three secretion system (T3SS), that are considered important in specific phases of *Shigella* pathogenesis [9].

GMMA have been proposed as OAg delivery systems. They are Outer Membrane Vesicles (OMVs) from Gram-negative bacteria genetically engineered to enhance their release [10] and to reduce their potential reactogenicity after injection, by modifying the lipid A structure, still maintaining the lipid A-triggered immunopotentiator effect of the TLR4 [11,12,13]. Rapid and simple manufacturing processes have been developed that allow obtaining high yields of GMMA, with the intent to produce low-cost vaccines [14]. GMMA faithfully resembles the composition of the bacterial outer membrane, and therefore, contains LPS, glycerophospholipids, outer membrane proteins, lipoproteins, and soluble periplasmic proteins in the lumen, triggering the activation of both innate and adaptive immunity. TLRs are of major interest in the recognition of GMMA-associated Pathogen-Associated Molecular Patterns (PAMPs) [11]. Among them, the most important TLRs for the recognition of PAMPs expressed on OMVs are TLR4 and TLR2. TLR4 is involved in the detection of the lipid A part of LPS [15] by forming a complex with the accessory proteins LPS Binding Protein (LBP), Myeloid Differentiation 2 (MD2), and cluster of differentiation 14 (CD14). TLR2 is directly involved in the recognition of bacterial lipoproteins through dimerization with TLR1 in the case of tri-acylated lipoproteins recognition, or TLR6 in the case of binding to bi-acylated lipoproteins [16].

LPS-specific humoral responses elicited by GMMA can be driven either by a T cell independent B-cell activation through LPS-driven BCR crosslinking or by a T cell-dependent pathway in which GMMA proteins are involved to activate helper T cells.

We have started with the development of a *S. sonnei* mono-valent GMMA-based vaccine adsorbed on Alhydrogel, demonstrated to be immunogenic in animals [13,17] and in healthy adults [18,19,20]. OAg-specific antibodies elicited by *S. sonnei* GMMA are functional in killing the bacteria in a complement-mediated fashion [21,22,23]. Additionally, anti-GMMA proteins antibodies are induced in mice upon immunization with S. *sonnei* GMMA [24].

*S. sonnei* GMMA have been combined with *S. flexneri* GMMA from three among the most prevalent serotypes (*S. flexneri* 1b, 2a and 3a) and adsorbed on Alhydrogel. The resulting quadrivalent GMMA-based vaccine is currently being tested in a phase I/II trial sponsored by GlaxoSmithKline Biologicals SA [25]. Alhydrogel was used as an adsorbent to further reduce the potential reactogenicity of GMMA particles [26].

In this study, we extended the work already performed to elucidate the contribution of *S. sonnei* GMMA proteins to GMMA immunogenicity [24], to *S. flexneri* GMMA. Furthermore, the role of GMMA proteins as T cell helpers and of TLR2 and TLR4 agonists present in GMMA on immunogenicity has been investigated to increase knowledge and better understand the potentialities of a GMMA-based vaccine against *Shigella*.

## 2. Results

### 2.1. GMMA Production and Characterization

*S. sonnei* GMMA displaying the OAg on their surface (OAg^+^) were produced from *S. sonnei* 53G strain (isolated from human beings with bacillary dysentery in Walter Reed Army Institute of Research*–*WRAIR). *S. flexneri* 1b OAg^+^ GMMA were produced from *S. flexneri* 1b NCTC5 STANSFIELD strain (Public Health England*–*PHE). *S. flexneri* 2a OAg^+^ GMMA were produced from *S. flexneri* 2a 2457T strain (received from WRAIR). *S. flexneri* 3a OAg^+^ GMMA were produced from *S. flexneri* 3a NCTC6885 strain (PHE). All strains were modified to increase the blebbing (ΔtolR), to attenuate their virulence by removing the virulence plasmid [13] and to modify the lipid A acylation status, thus obtaining strains carrying lipid A with reduced endotoxicity [12]. Matrix-assisted laser desorption/ionization-time of flight (MALDI-TOF) mass spectrometry analysis has been used to confirm modified lipid A acylation status in the resulting mutants, with respect to the corresponding wild type strains (Appendix A). GMMA not displaying the OAg on their surface (OAg^−^) were produced from the relative OAg^+^ *S. flexneri* strains through the introduction of Δ*rfbG mutation* (Table 1).

OAg and proteins content of all GMMA preparations are reported in Table 1. The OAg/protein ratio was similar for all OAg-positive GMMA but those of *S. sonnei* were characterized by lower OAg density. The expected sugar composition was confirmed by high-performance anion exchange chromatography–pulsed amperometric detection (HPAEC–PAD). The same analysis confirmed the absence of OAg in OAg-negative samples. OAg chain length was estimated by size exclusion-high-performance liquid chromatography (HPLC-SEC) analysis using dextrans as standards. All OAg*^+^* GMMA have a main OAg peak in the range of 14–19 kDa. *S. sonnei* GMMA also produces a capsular polysaccharide at a higher molecular weight (~220 KDa) and *S. flexneri* 2a GMMA have an additional OAg population of approximately 50 kDa [27]. The GMMA protein pattern was recorded by reverse phase-high-performance liquid chromatograph (HPLC-RP) (Appendix A). Similar protein patterns were observed for all GMMA samples.

The global set of proteins present in our GMMA was characterized by performing a proteomic analysis based on liquid chromatography with tandem mass spectrometry (LC-MS/MS). By focusing on proteins having a relative abundance > 0.5% (*w*/*w*) in each one of the 6 S. flexneri GMMA preparations, which accounted for >80% of the total protein composition of each sample, we obtained a final list of 20 proteins (Figure 1).

The relative abundance of some proteins was significantly different between OAg-positive and OAg-negative GMMA of each *S. flexneri* strain (Appendix A) and more differences were evidenced in the comparison among the three OAg-positive or OAg-negative GMMA generated from the three *S. flexneri* strains (Appendix A), with some proteins not quantified in certain samples as well. However, in all GMMA samples, the most represented proteins were OmpA and OmpC, which account for ~50% (*w*/*w*) of the total protein content.

### 2.2. Role of GMMA Proteins in Vaccine-Elicited Immune Response

*S. flexneri* OAg-positive and OAg-negative GMMA were compared in mice at the same dose of GMMA proteins of 10 µg to maximize the immune response against GMMA proteins. GMMA were administered intraperitoneally since in a preliminary study this immunization route resulted in a stronger anti-GMMA proteins response compared to subcutaneous or intramuscular routes. The OAg density in the OAg-positive GMMA was similar among the three samples and, as a consequence, the groups of mice immunized with OAg-positive GMMA received a comparable dose of OAg (≈8 µg).

OAg-specific and GMMA proteins-specific total IgG elicited by vaccination with OAg-positive and OAg-negative GMMA were quantified through enzyme-linked immunosorbent assay (ELISA), while sera functionality was expressed as the half maximal inhibitory concentration (IC_50_) and assessed against *S. flexneri* 1b, 2a and 3a wild type strains through a luminescent based serum bactericidal assay (SBA) and an opsonophagocytic killing assay (OPK).

*S. flexneri* OAg-positive GMMA elicited significant OAg-specific total IgG levels already 27 days after the first vaccination, which increased 14 days after the second injection. Even though a low signal was observable after immunization with OAg-negative GMMA, the total IgG response was significantly lower compared to that elicited by the corresponding OAg-positive GMMA, maybe due to core-specific antibodies (Figure 2a,b). For *S. flexneri* 3a GMMA at day 27, the OAg-specific IgG response elicited by OAg-positive and OAg-negative GMMA was similar but pretty low.

Immunization with both OAg-positive and OAg-negative GMMA elicited the anti-GMMA proteins’ total IgG. Antibody levels were higher after immunization with OAg-negative GMMA both after prime and boost for *S. flexneri* 2a and 3a GMMA, not the same for *S. flexneri* 1b, for which OAg-positive and OAg-negative GMMA elicited a not significantly different anti-GMMA proteins IgG response (Figure 2c,d).

Only antibodies elicited after immunization with OAg-positive GMMA mediated killing of the corresponding OAg-positive *S. flexneri* strains in SBA, whereas antibodies elicited after immunization with OAg-negative GMMA did not (Figure 2e).

Similarly, antibodies elicited after immunization with OAg-positive GMMA mediated opsonophagocytosis of the corresponding OAg-positive *S. flexneri* strains. Antibodies elicited after immunization with *S. flexneri* 1b and 2a OAg-negative GMMA did not show any activity in OPK against the corresponding OAg-positive *S. flexneri* strains. Antibodies elicited after immunization with *S. flexneri* 3a OAg-negative GMMA showed low but detectable opsonophagocytic activity against the corresponding strain (Figure 2f).

In addition, GMMA-associated proteins may stimulate the cooperation of T-helper cells with B cells for induction of a T cell-dependent OAg-specific response. With the purpose to investigate this feature, wild type and T cell-deficient mice were immunized with *S. sonnei* or *S. flexneri* 2a GMMA and the OAg-specific humoral response was quantified. Twenty-seven days after the first injection, GMMA elicited a significant IgG response in T cell-deficient mice, but it was much lower than in the wild type mice, highlighting a strong T-dependent component of the immune response induced by GMMA (Figure 3a,b). Consistent with a T-dependent humoral response, two weeks after the second injection, the antibody production was boosted in wild type animals, whereas it did not increase in T cell-deficient mice (Figure 3a,b). In line with the antibody production, the bactericidal activity of the sera collected after the second vaccination was significantly reduced in the T cell-deficient mice, in particular against *S. flexneri* 2a, for which many non-responder animals were found among T cell-deficient mice (Figure 3c,d)

### 2.3. Influence of TLR2 and TLR4 Agonists on the Humoral Response Elicited by the Quadrivalent GMMA-Based Vaccine against Shigella

A quadrivalent GMMA-based vaccine formulated with Alhydrogel is currently being tested in a phase I/II trial [25]. With the intent of elucidating the role of TLR4 and TLR2 agonists present in the quadrivalent GMMA-based vaccine in inducing immunogenicity, TLR4 mutant and TLR2 KO and corresponding wild type mice were immunized with the quadrivalent GMMA formulation. Alhydrogel was used in the vaccine as an adsorbent to further reduce the potential reactogenicity of the GMMA particles [26] and not as an adjuvant. Therefore, in this set of experiments, we compared the immunogenicity elicited by the quadrivalent GMMA formulation either in the presence or absence of Alhydrogel. With the purpose to mimic the immunization route used in the clinical trials mice have been immunized intramuscularly. The GMMA dose has been decreased to 9.4 ng of each one of the four GMMA (quantified as OAg) to see differences not being at the plateau of the anti-OAg response.

In the absence of Alhydrogel, we observed a significant reduction in the serum bactericidal activity against all four *Shigella* species whose reference GMMA are contained in the quadrivalent vaccine (Figure 4b). This negatively affected bactericidal activity was associated with a significantly reduced OAg-specific IgG response, with the exception of *S. flexneri* 1b (Figure 4a).

In the presence of Alhydrogel, OAg-specific IgG response was significantly higher in wild type mice than in TLR4 mutant mice only for *S. flexneri* 1b and 2a GMMA. Instead, SBA titers were significantly higher in wild type mice than in TLR4 mutant mice after vaccination with *S. flexneri* 2a and 3a GMMA. The presence of Alhydrogel reduced the functional responses elicited by *S. flexneri* 1b and 3a GMMA.

Interestingly, no impact of impaired TLR4 signaling was observed on the magnitude of GMMA proteins-specific humoral response (Figure 5a).

No statistically significant difference was observed between the wild type and TLR2-deficient mice in the levels of OAg-specific antibodies and serum bactericidal activity elicited by the quadrivalent GMMA formulation either in the absence or in presence of Alhydrogel (Figure 6). No impact of impaired TLR2 signaling was observed on the magnitude of GMMA proteins-specific humoral response as well (Figure 5b).

## 3. Discussion

GMMA have been proposed as an attractive platform for the development of a potential low-cost *Shigella* vaccine, eliciting broad protection against the most prevalent *Shigella* serotypes [25]. We have recently shown that *S. sonnei* GMMA can elicit not only LPS-specific but also GMMA proteins-specific antibodies. However, the presence of OAg chains on the bacterial surface shields the bacteria from anti-protein antibody binding, and therefore, anti-OAg antibodies are the main drivers of bactericidal activity against OAg-positive bacteria. This finding was reinforced by the functional analysis of human sera from a phase 2b study of the *S. sonnei* GMMA-based vaccine [23]. The adsorption of anti-OAg antibodies from post-vaccination sera confirmed that anti-protein antibodies were not able to induce complement-mediated bactericidal killing against *S. sonnei* OAg-positive bacteria. Interestingly, antibodies not targeting the OAg are functional against OAg-negative bacteria, and immunodominant protein antigens were identified by proteomic analysis [24].

Here, in light of the development of a quadrivalent vaccine against *Shigella*, we extended the study performed with *S. sonnei* GMMA to *S. flexneri* 1b, 2a and 3a GMMA. Similarly to what conducted for *S. sonnei*, *S. flexneri* GMMA were mutated in order not to display the O-antigen on their surface to better elucidate the contribution of proteins to *S. flexneri* GMMA immunogenicity.

Through proteomic analysis, we found that protein composition is similar among the different *S. flexneri* GMMA samples tested, both OAg-positive and OAg-negative, despite some differences in their relative expression level. The most abundant outer membrane proteins in the GMMA were identified as OmpA and OmpC, consistent with previous observations with *S. sonnei* and *S. flexneri* 2a GMMA [10,24,28] and with the preparation of GMMA from cultures in the early exponential phase [29]. T3SS proteins (e.g., Ipa) cannot be present in GMMA, as a consequence of virulence plasmid removal in GMMA producer strains.

When tested in mice, *S. flexneri* 1b, 2a and 3a OAg-positive GMMA were able to elicit higher anti-OAg specific antibody titers compared to OAg-negative GMMA, as expected. Anti-OAg total IgG elicited after immunization with OAg-negative GMMA was probably due to the presence of the core on the surface of OAg-negative GMMA as well as in the ELISA plate coating antigen. On the other hand, despite the same protein dose being used for OAg-positive and OAg-negative GMMA, OAg-positive GMMA elicited lower anti-protein responses, but for *S. flexneri* 1b GMMA, suggesting an immuno-interference between OAg and protein antigens. Reasons for the different behavior of *S. flexneri* 1b GMMA could be better investigated, for example, by looking at the OAg chain length on the different GMMA, whereas the OAg/protein ratio was similar among the three GMMA samples. Overall, results obtained with *S. flexneri* GMMA were, therefore, in accordance with what was observed for *S. sonnei* GMMA [24].

Although an immune correlate of protection against shigellosis has not yet been established, two well-characterized assays are available to measure the functionality of the vaccine-induced antibodies against *Shigella* spp. SBA and OPK [30]. Even though both OAg-positive and OAg-negative GMMA were able to elicit anti-protein antibodies, sera elicited against OAg-positive GMMA were able to kill the corresponding *S. flexneri* in SBA, whereas sera raised against OAg-negative GMMA did not show any complement-mediated bactericidal activity on the corresponding OAg-positive *S. flexneri* strain. Similar results were obtained by OPK, where only sera raised against OAg-positive GMMA showed opsonophagocytic activity, whereas sera raised against OAg-negative GMMA did not. A very low signal in OPK was observed only with sera from mice immunized with *S. flexneri* 3a OAg-negative GMMA, but the titers were very low compared to those elicited by the corresponding OAg-positive GMMA. The observed results reinforced previous findings reported for *S. sonnei* GMMA [24] that OAg is indeed a key antigen for functional immunity against *Shigella* strains, but antibodies against protein antigens are generated anyway upon immunization and may be an added value of GMMA vaccines since *Shigella* pathogenesis is complex [31] and antibodies against targets other than OAg might be important in protecting against the spread of the infection in specific organs or time of infection.

Moreover, GMMA-associated proteins were demonstrated to stimulate the cooperation of T-helper cells with B cells for the induction of a T cell-dependent vaccine-specific humoral response. Indeed, GMMA induced a much stronger LPS-specific humoral response in wild type than in T cell-deficient mice. These results confirm that GMMA are not only a delivery system for *Shigella* OAg since they also display additional protein antigens which are strongly immunogenic and engage T cells for cooperation with B cells to elicit a T cell-dependent humoral response [27,32,33,34]. On the other hand, the LPS-specific IgG observed in T cell-deficient mice represents the T-independent component of humoral response elicited by GMMA.

GMMA also stimulate innate immune cells [11]. Indeed, GMMA induce a pro-inflammatory immune response in human peripheral blood mononuclear cells (PBMSCs). Production of IFN-γ, IL-10, IL-12p70, IL-1β, IL-6, TNF-α, IL-8HA. IL-2, IL-4, Eotaxin, MCP-1, MIP-1α, MIP-1β, GM-CSF, IL-12p40, and IL-1α was observed after PBMC stimulation [12,35]. TLRs are engaged in the recognition of GMMA components, mainly TLR4 and TLR2 [12] and the TLR agonists present on GMMA may serve as immunopotentiators for the quadrivalent GMMA-based vaccine. Indeed, several synthetic TLR agonists have been selected as vaccine adjuvants and have been tested in clinical trials [36].

When the four-component GMMA formulation was tested in absence of Alhydrogel in mice with impaired TLR4 signaling, the levels of vaccine-specific functional total IgG were reduced in comparison to those observed in wild type mice.

When Alhydrogel was added to the formulation with the unique purpose of further reducing GMMA potential endotoxicity, an impact on the humoral response was observed only for *S. flexneri* 2a OAg-specific antibodies. This is probably due to the fact that, in the presence of Alhydrogel, the humoral response in wild type mice is already reduced compared to that elicited by unadjuvanted GMMA. It would be useful to further explore the role that the adsorption of GMMA on Alhydrogel can have on the immune response elicited by GMMA. Interestingly, the adjuvanticity effect of TLR4 agonists on GMMA is exerted only on OAg-specific but not protein-specific humoral response in the experimental conditions tested in this study. On the other hand, TLR2 agonists present in GMMA are dispensable for the vaccine-elicited humoral response. Such results are in line with previous in vivo studies performed with *S. sonnei* and *Salmonella* Typhimurium GMMA [37] showing that TLR4 engagement plays a substantial role in inducing antibody production, and its role can be obscured by the effect of Alhydrogel, whereas TLR2 engagement has no effect. Similar results have been also generated in mice after immunization with *Neisseria meningitidis* detergent-extracted OMV [38].

This study showed that, when tested in the animal model, GMMA combines both adjuvant and carrier activities, which allows for an increase in the low immunogenic properties of carbohydrate antigens alone. Results obtained in mice will be soon corroborated by results from ongoing clinical studies, and are important to better understand how GMMA work and to design optimal vaccines based on this technology.

## 4. Materials and Methods

### 4.1. Bacterial Strains and Generation of Mutants

*S. flexneri* 1b and 3a were acquired from Public Health England (PHE, London, UK) whereas *S. sonnei* and *S. flexneri* 2a were received by WRAIR (Silver Spring, MD, USA). All strains were engineered to obtain different mutants. The null mutations were obtained by replacing the genes of interest with an antibiotic resistance cassette by homologous recombination using the lambda red recombineering system [39]. In some cases, removal of the antibiotic selective marker was performed after gene deletion using the plasmid pCP20. In the case of *S. sonnei*, the virulence plasmid-encoded *virG* gene was replaced with the *nadA* and *nadB* genes from *E. coli*, as described by Gerke et al. [13]. For all *S. flexneri* strains, white colonies were selected on Congo red agar, indicating the loss of the virulence plasmid pINV. The *tolR* gene was replaced with the kanamycin resistance gene aph, as described by Berlanda Scorza et al. [10]. The *msbB* genes were replaced with the erythromycin and chloramphenicol resistance genes *erm* and *cat*, as described by Rossi et al. [12] and Mancini et al. [24]. Finally, the *rfbG* gene (essential for OAg biosynthesis) was replaced with the erythromycin gene *erm* as described by Rossi et al. [12].

### 4.2. GMMA Production and Characterization

*Shigella* GMMA were produced and purified as previously described [13]. Briefly, cultures of the different GMMA-producing strains were diluted in Chemically Defined Medium (see Appendix A) to an optical density at 600 nm (OD600) of 0.3 and grown overnight at 30 °C in baffled flasks with a liquid-to-air volume ratio of 1:5.

Total protein content was estimated by microBCA using bovine serum albumin (BSA) as a reference following the manufacturer’s instructions (Thermo Scientific, Waltham, MA, USA), while total OAg amount was determined by HPAEC-PAD [40,41], extracted OAg was characterized by HPLC-SEC [41,42], lipid A structure was determined by MALDI-TOF mass spectrometry [12,41]. For the OAg-negative GMMA, the absence of OAg repeats was confirmed by HPAEC-PAD analysis, with a lack of the characteristic sugars of the OAg repeating unit.

GMMA protein pattern was recorded by HPLC-RP using an Acquity H-class (Waters, Milford, MA, USA) equipped with a fluorimeter (ex 280*/*em 336 nm) using an Acquity Protein BEH C4 column 300 Å 1.7 µm 2.1 *×* 100 mm (Waters 186004496) and the eluent program reported in Appendix A.

### 4.3. Proteomic Analysis

For quantitative proteomic analysis, GMMA samples were prepared as previously described [24]. For each GMMA sample, two (technical replicates) tryptic digestions were performed followed by single LC–MS/MS acquisitions as previously described [24]. The percentage of each protein in the total sample was calculated according to the corresponding peak area (averaged between the two technical replicates) and the theoretical molecular weight (MW) using the following formula:%ProteinX = AvgAreaProteinX × MWProteinX/∑(AvgAreaProtein × MWProtein)

The mass spectrometry raw data were processed with the PEAKS software ver. 8 (Bioinformatics Solutions Inc.) for peptide sequence assignment, de novo sequencing, database matching identification and label-free quantification, as previously described [43]. Protein identification from MS/MS spectra was performed against *S. flexneri* 2a str. 2457T database (NCBI code GCF_000183785.1_ASM18378v2_protein.faa, 3984 ORF entries) combined with common contaminants (human keratins and autoproteolytic fragments of trypsin) with a FDR set at 0.1%. From the analysis, only proteins with relative abundance ≥ 0.5% *w*/*w* were reported.

### 4.4. Mouse Studies

All animal studies were performed at GSK Animal Resources Centre under the animal project 479/2017-PR, 527/2020-PR and 471/2020-PR approved by the Italian Ministry of Health. For evaluation of immunogenicity of *S. flexneri* OAg-positive and OAg-negative GMMA, groups of eight CD1 mice (female, 5 weeks old) were vaccinated intraperitoneally with 10 µg of GMMA (protein dose based on microBCA quantification) in 200 µL of saline at study day 0 and 28. For evaluation of the GMMA proteins’ impact on GMMA immunogenicity, groups of eight CD1 mice and T cell-deficient mice (Crl:CD1-Foxn1^nu^) were immunized subcutaneously (as in previous experiments [33]) at days 0 and 28 with 0.5 µg (quantified as total OAg) of *S. sonnei* and *S. flexneri* 2a GMMA. For evaluation of the impact of TLR4 and TLR2 agonists on GMMA immunogenicity, groups of 10 female, 5 weeks old BALB/cAnN or TLR2^−/−^ or TLR4^mut^ (C3H mice) were vaccinated intramuscularly with the quadrivalent GMMA formulation at study day 0 and 28: 9.4 ng each OAg in 50 µL were administered to each mouse, either in absence or presence of 0.7 mg/mL of Alhydrogel (Al^3+^). Serum was collected on days 1 (pooled sera), 27 and 42 (individual sera).

### 4.5. Ethics and 3Rs Statement

All animal experiments were performed in accordance with relevant national and international legislation (Italian Legislative Decree 26/2014 and European Directive for the Use of Animals for Scientific Purposes 2010/63) and GSK Animal Welfare Policy and standards. All animal protocols were reviewed by the local Animal Welfare Body and approved by the Ministry of Health, according to the above-mentioned legislation.

GSK is committed to the replacement, reduction and refinement of animal studies (3Rs). Non-animal models and alternative technologies are part of our strategy and are employed where possible. When animals are required, the application of robust study design principles and peer review minimizes animal use, reduces harm and improves benefit in studies.

### 4.6. Assessment of Anti-Shigella OAg and Anti- GMMA Proteins Immune Responses in Mice

Pre-immune sera and sera collected four weeks after the first and two weeks after the second immunization were analyzed by ELISA [24] for anti-*S. sonnei* OAg total IgG content using *S. sonnei* LPS as plate coating antigen (at the concentration of 0.5 µg/mL in Phosphate Buffer Saline, PBS), for anti-*S. flexneri* 1b OAg total IgG content using *S. flexneri* 1b OAg as plate coating antigen (at the concentration of 2 µg/mL in Carbonate Buffer), for anti-*S. flexneri* 2a OAg total IgG content using *S. flexneri* 2a OAg as plate coating antigen (at the concentration of 5 µg/mL in Carbonate Buffer), for anti-*S. flexneri* 3a OAg total IgG content using *S. flexneri* 3a OAg as plate coating antigen (at the concentration of 1 µg/mL in PBS) and for anti-*Shigella* GMMA proteins total IgG content using *S. flexneri* 2a OAg- GMMA as plate coating antigen (at the concentration of 15 µg/mL in PBS). Results are expressed in ELISA units (EU/mL) determined relative to a standard serum curve. One ELISA unit equals the reciprocal of the dilution of the standard serum giving an OD405nm-490nm of 1 in the assay.

### 4.7. Assessment of Serum Bactericidal Activity against Shigella

Single sera collected at day 42 were assayed in SBA based on luminescent readout as previously described [44] against OAg-positive *S. sonnei*, *S. flexneri* 1b, 2a and 3a [45]. Heat-inactivated (HI) sera were serially diluted in PBS in the SBA plate (25 µL/well). The starting dilution of each serum in the assay was 1:100 (final dilution), followed by three-fold dilution steps up to seven dilution points, plus one control well with no sera added. A four-parameter non-linear regression was applied to raw luminescence (no normalization of data was applied) obtained for all the sera dilutions tested for each serum; an arbitrary serum dilution of 10^15^ was assigned to the well containing no sera. Fitting was performed by weighting the data for the inverse of luminescence^2 and using GraphPad Prism ver. 7 software (GraphPad Software).

Results of the assay are expressed as the IC*_50_*, represented by the reciprocal serum dilution that is able to reduce the luminescence signal by 50% compared to the negative control (and thus causes 50% growth inhibition). Titers lower than the minimum measurable assayed were assigned a value of half of the first dilution of sera tested (50).

### 4.8. Assessment of Serum Opsonophagocitic Activity against Shigella

OPK performed on sera collected at day 42 against OAg-positive *S. flexneri* 1b, 2a and 3a are based on CFU counts and was adapted from Ramachandran et al. [46]. Heat-inactivated (HI) sera were serially diluted in OPB (Opsonization buffer) in the OPA plate at the appropriate dilution for each of tested strain and then up to seven three-fold dilutions were prepared; a control well with no sera was added. Sera were incubated with 300 colony-forming units (CFU) of tested bacteria culture for 30 min at 37 °C with 5% CO_2_. Then 4 × 10^5^ dimethylformamide differentiated HL-60 cells were added to opsonized bacteria with 2.5% of Baby Rabbit Complement (Cederlane, Burlington, Ontario, Canada) in a final volume of 100 µL. OPA reaction was incubated for 90 min at 37 °C with 5% CO_2_ and kept on ice for 20 min to halt the phagocytic process; 10 µL of OPA reaction were plated and tilted on LB agar plates and incubated O/N at 37 °C. CFU counts were used to calculate the IC*_50_*, corresponding to the serum dilution needed to reduce the CFU number by 50% compared to the negative control and were calculated by applying a four-parameter non-linear regression (no normalization of data was applied). Fitting was performed by weighting the data for the inverse of luminescence^2 and using GraphPad Prism ver. 7 software (GraphPad Software).

### 4.9. Statistical Analysis

Significant differences between ELISA responses, SBA or OPK titers were evaluated using the non-parametric two-tailed Mann–Whitney test.

## Figures and Tables

**Figure 1 ijms-24-02742-f001:**
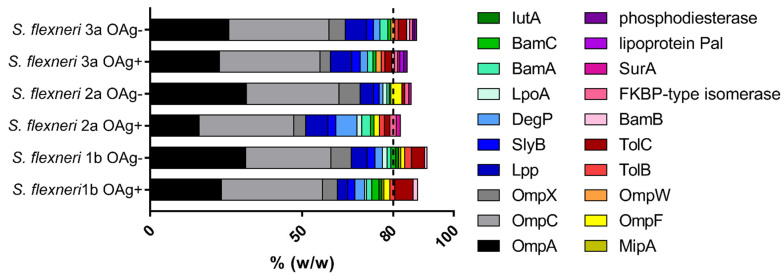
Proteomic analysis of *S. flexneri* GMMA (most abundant proteins). Proteomic analysis was performed by LC–MS/MS. A global set of 20 proteins was identified as those showing an abundance superior to 0.5% in the six different samples. The abundance of each protein in the OAg-positive and OAg-negative GMMA was compared. Results derived from two independent trypsin digestions of each GMMA preparation, each of them acquired in a single LC-MS/MS acquisition.

**Figure 2 ijms-24-02742-f002:**
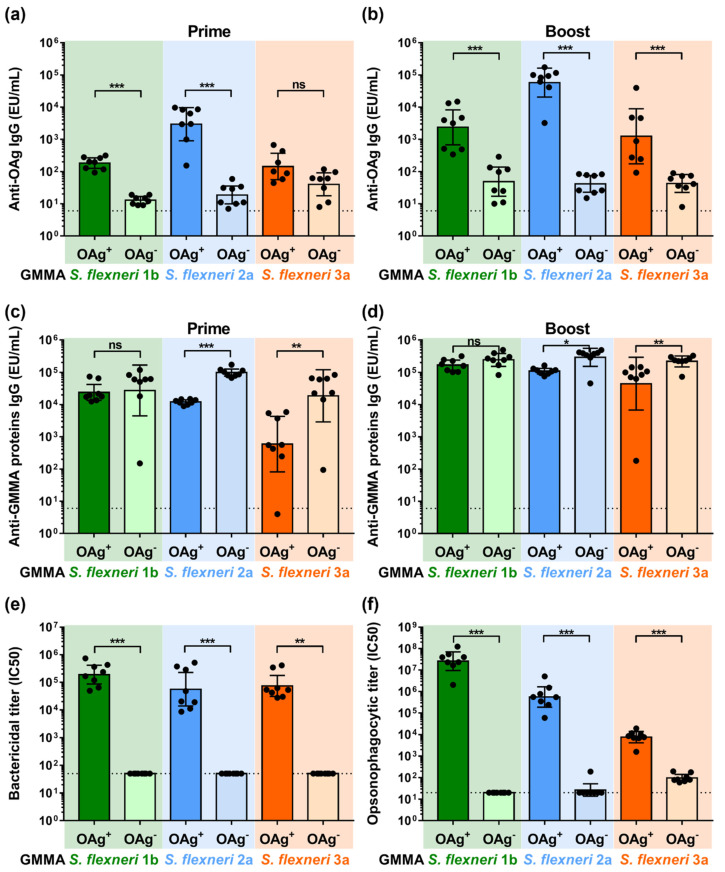
Characterization of the humoral response elicited by OAg^+^ and OAg^−^ *S. flexneri* GMMA in mice immunized intraperitoneally at day 0 and 28 with 10 µg of total GMMA proteins. (**a**) *S. flexneri* OAg-specific total IgG (EU/mL) after prime; (**b**) *S. flexneri* OAg-specific total IgG (EU/mL) after boost; (**c**) *S. flexneri* GMMA proteins-specific total IgG (EU/mL) after prime; (**d**) *S. flexneri* GMMA proteins-specific total IgG (EU/mL) after boost. (**e**) Bactericidal and (**f**) opsonophagocytic titers after boost expressed as IC_50_. Geometric mean (bar) is reported for all groups together with individual values (dots) and 95% confidence intervals (CI). Dotted lines represent titer of pre-immune sera in panel (**a**–**d**), titer of not bactericidal sera in panel (**e**) and titer of not opsonizing sera in panel (**f**). * *p* < 0.05; ** *p* < 0.01; *** *p* < 0.001; ns *p >* 0.05, Mann–Whitney test.

**Figure 3 ijms-24-02742-f003:**
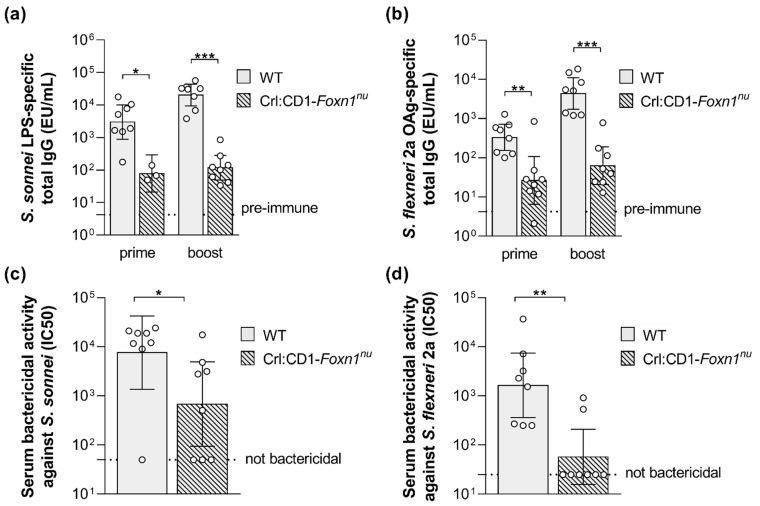
*S. sonnei* and *S. flexneri* GMMA compared in wild type and T cell-deficient mice. *S. sonnei* LPS-specific (**a**) and *S. flexneri* 2a OAg-specific (**b**) total IgG (EU/mL) after prime and boost in wild type (WT) and T cell-deficient mice (Crl:CD1-*Foxn1^nu^*) immunized subcutaneously at day 0 and 28 with 0.5 µg (quantified as total OAg) of GMMA. Bactericidal titers against *S. sonnei* (**c**) and *S. flexneri* 2a (**d**) after boost expressed as IC_50_. Geometric mean (bar) is reported for all groups together with individual values (dots) and 95% confidence intervals (CI). * *p* < 0.05; ** *p* < 0.01; *** *p* < 0.001, Mann–Whitney test.

**Figure 4 ijms-24-02742-f004:**
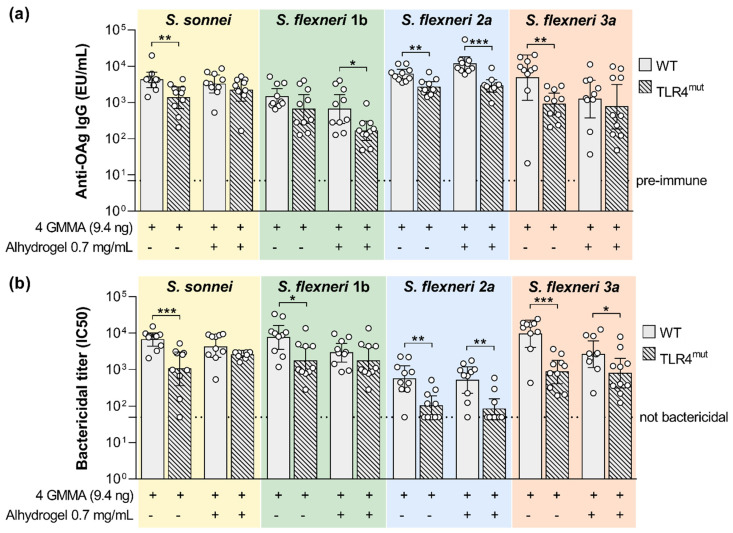
Impact of TLR4 agonists present on GMMA on the polysaccharide-specific humoral response elicited by *Shigella* GMMA-based vaccine. (**a**) LPS-specific (*S. sonnei*) or OAg-specific (*S. flexneri*) total IgG (EU/mL) after boost observed in wild type (WT) and TLR4 mutant mice (TLR4^mut^). (**b**) Bactericidal titers expressed as IC_50_. Mice were immunized intramuscularly at days 0 and 28 with the quadrivalent GMMA-based vaccine against *Shigella*: 9.4 ng OAg of each one of the four GMMA have been administered to each mouse, either in absence or presence of 0.7 mg/mL of Alhydrogel Al^3+^. Geometric mean (bar) is reported for all groups together with individual values (dots) and 95% confidence intervals (CI). Responses elicited in WT and TLR4^mut^ mice by each GMMA were compared by Mann–Whitney test; * *p* < 0.05; ** *p* < 0.01; *** *p* < 0.001.

**Figure 5 ijms-24-02742-f005:**
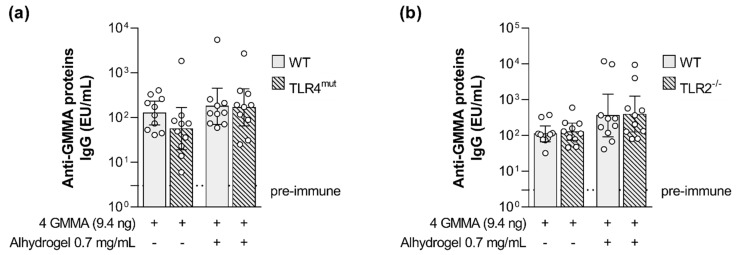
Impact of TLR4 and TLR2 agonists present on GMMA on the GMMA proteins-specific humoral response elicited by *Shigella* GMMA-based vaccine. (**a**) *Shigella* GMMA proteins-specific total IgG (EU/mL) after boost observed in wild type and TLR4 mutant (TLR4^mut^) mice and in (**b**) wild type and TLR2 deficient (TLR2^−/−^) mice. Mice were immunized intramuscularly at days 0 and 28 with the quadrivalent GMMA-based vaccine against *Shigella*: 9.4 ng OAg of each one of the four GMMA have been administered to each mouse, either in absence or presence of 0.7 mg/mL of Alhydrogel (Al^3+^). Geometric mean (bar) is reported for all groups together with individual values (dots) and 95% confidence intervals (CI). No statistically significant difference was observed between WT and TLR4^mut^ or WT and TLR2^−/−^ mice using Mann–Whitney test.

**Figure 6 ijms-24-02742-f006:**
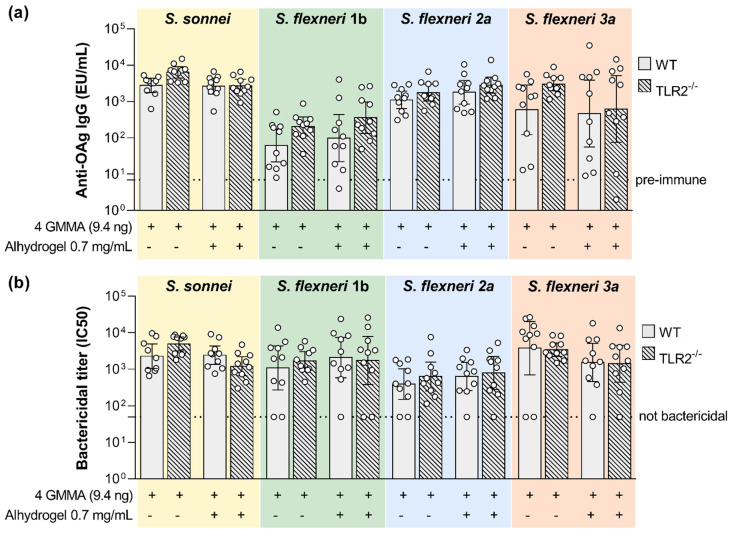
Impact of TLR2 agonists present on GMMA on the polysaccharide-specific humoral response elicited by *Shigella* GMMA-based vaccine. (**a**) LPS-specific (*S. sonnei*) or Oag-specific (*S. flexneri*) total IgG (EU/mL) after boost observed in wild type (WT) and TLR2 deficient mice (TLR2^−/−^). (**b**) Bactericidal titers expressed as IC_50_. Mice were immunized intramuscularly at days 0 and 28 with the quadrivalent GMMA-based vaccine against *Shigella*: 9.4 ng (quantified as total Oag) of each one of the four GMMA have been administered to each mouse, either in absence or presence of 0.7 mg/mL of Alhydrogel. Geometric mean (bar) is reported for all groups together with individual values (dots) and 95% confidence intervals (CI). No statistically significant difference was observed between WT and TLR2^−/−^ mice using Mann–Whitney test.

**Table 1 ijms-24-02742-t001:** GMMA used in the study. The table summarizes all GMMA main characteristics and relative GMMA producer bacterial strains. n.d.: not determined.

Antigen	GMMA Producing Strain	OAg Concentration (µg/mL)	Protein Concentration (µg/mL)	OAg/Protein Ratio
*S. sonnei* OAg^+^ GMMA	*S. sonnei virG::nadAB, tolR::aph msbB1::erm msbB2::cat*	537.4	1858.9	0.29
*S. flexneri* 1b OAg^+^ GMMA	*S. flexneri* 1b Δ*tolR::frt msbB1a::frt msbB1b::frt*	2179.4	2264.9	0.96
*S. flexneri* 1b OAg^−^ GMMA	*S. flexneri* 1b Δ*tolR::frt msbB1a::frt msbB1b::frt rfbG::erm*	0 (no OAg detected)	1339.3	n.d.
*S. flexneri* 2a OAg^+^ GMMA	*S. flexneri* 2a Δ*tolR::aph msbB::cat*	1400.9	1794.6	0.78
*S. flexneri* 2a OAg^−^ GMMA	*S. flexneri* 2a Δ*tolR::aph msbB::cat rfbG::erm*	0 (no OAg detected)	1814.6	n.d.
*S. flexneri* 3a OAg^+^ GMMA	*S. flexneri* 3a *tolR::aph msbB::cat*	1961.1	1820.6	1.07
*S. flexneri* 3a OAg^−^ GMMA	*S. flexneri* 3a *tolR::aph msbB::cat rfbG::erm*	0 (no OAg detected)	784.5	n.d.

## Data Availability

Not applicable.

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
