# Peer review of "Exploring the Role of GMMA Components in the Immunogenicity of a 4-Valent Vaccine against Shigella"

_ijms, 2023, doi:10.3390/ijms24032742_

Round 1

Reviewer 1 Report

It is interesting extension of work from the past based on S. Sonnei GMMA and takes a leap toward 4 strain combined vaccine. However, few points may need to be addressed before it can be published. Especially teh use of GMMA is intersting as it provides antigen and adjuvant into one component

1. Provide some reference related to other vaccine research for Shigella antigen and/or adjuvants including oral vaccine developed using Shigella dysenteriae O-specific polysaccharide vaccine

2. What is the need for usig Alhydrogel in the immunization experiment is not clear.

3. Is there any ex-vivo cytokine induction data available?

4. Please discuss the toxicity of GMMA which is LPS component. Since mice is less senstive to LPS, did you use galactoseamine or high fat/cholesterol diet to increase the sensitivity to LPS in the immunization experiment. 

5. What is the units for total IgG antibodies (EU/mL) mean? Usually EU/mL is referred to endotoxin units/mL.

6. Does the presence of O-Ag affect the peptide characterization by LC-MS/MS and whether removal of all carbohydrate units considered to see teh difference between protein composition?

Other minor corrections include sentence and spelling corrections at teh following locations:

Page 1, line 23-25

Page 2, line 63

Page 3, line 102

Page 3, line 110, correct eth spelling of abundance

Page 4, line 150, change shown to show

Page 9, line 250 change contribute to contribution

Change IC50 to include '50' as subscript as in IC50

Reviewer 2 Report

The data presented in the manuscript are informative and provide enough information to make some basic conclusions regarding the S. flexneri GMMAs.  Characterization of the S. flexneri GMMA preparations appears appropriate, but there’s a paucity of specific information regarding the OAg.  For example what is the average OAg repeat length?  What is the state of state of lipid A acylation?  The experiments designed to determine the “role of GMMA proteins in vaccine-elicited immune responses” fall far short of this goal and provide little insight on how these proteins contribute to vaccine immunogenicity.  For example, the SBA/OPK experiments use target strains that contain OAg; however, a more direct way to assess the role of GMMA proteins would be to use strains that lack the OAg.  Again, this type of experimental set up would be a more direct way to assess the role anti-protein responses to bactericidal activity.  Similarly, the experiments involving T-cell deficient mice don’t address the role of protein GMMA antigens at all.  In general, the authors should prepare GMMAs from strains lacking the key OMPs and use this to conduct immunogenicity and SBA experiments.  The experiments looking at the role of TLR2/TLR4 are presenting in a very confusing manner and the rationale for using the 4 valent GMMA these experiments are not explained.  Also, the purpose of using alyhydrogel is not explained nor justified.  Why is the overall immunogenicity of the 4-valent GMMA seemingly lower in these experiments?  Why do these experiments use a different route of administration for vaccination (IP vs IM)?  This section of the manuscript (i.e., section 2.3) needs to be re-worked to include rationale for the experimental design and to include proper comparisons across the entire dataset.  

As presented, the experiments in the manuscript fail to provide a cohesive compendium and don’t significantly advance our understanding of the Shigella GMMA vaccine platform.  

Reviewer 3 Report

The manuscript of Mancini et al. describes the possible use of GMMA components combined with 3 prevalent S. flexneri serotypes to promote a humoral response against OAg and proteins of the GMMA types used. They demonstrate that GMMA, in mice, have good adjuvant and carrier activities increasing the low immunogenic properties of the carbohydrate antigens. They have also shown that the humoral response is T-cell dependent.

Major comments:

-       Page 2, line 81 to 88: Please review this phrase, to long and confusing. “which were modified to …”. It is not clear what is the strain that is modified, or if each modification belongs to different strains.

-       Table S1 to S5: The CV abbreviation is for “coefficient of variation”? Please describe this abbreviation in the table legend.

-       Table S1 to S5: If CV is “coefficient of variation” you have several higher than 30%, can you please justify this high variability in the values used and if interferes with the analyses performed.

-       Page 4, line 116 to 118: “No major differences were observed ….”. After the analysis of figure 1 and tables S1 to S5 I considered that exist some differences between the abundances of same proteins detected, however I don’t know the statistical significance of these differences? (Examples: OmpA abundance decreases from S. flexneri 1b or 2a OAg- to OAg+; DegP increases from S. flexneri 2a OAg- to OAg+). Several proteins are not detected in all the GMMA samples, suggesting also differences in the proteins detected (Table S4 and S5).

-       Page 4, line 118 to 120: OmpA and OmpC are the most represented, however in some samples the OmpX is more abundant and in others is Lpp, so I suggest adding the Lpp in the most abundant also. Or only include the OmpA and OmpC in this phrase.

-       Figure 2, 3 and 4: The route of immunization was different in each mice experiment performed (intraperitoneal, subcutaneous and intramuscular). Can the authors please explain the reason for not using only one common route of immunization in all assays?

Minor comments:

-       Table S1 to S5: The tables have same gaps in the protein that were not detected. I suggest changing to “nd” (not detected) instead of the gap, will be more clear to the reader. Add this information to the table legend.

-       Page 7, line 187 to 190: Please review this phrase, to long and confusing.

-       Page 11, line 343-344: “Chemically Defined Medium” is missing composition or if commercial please add the company information.

-       Please review the reference list. Some references incomplete and others with small typing errors (ref. 3, 4, 23, 27, 30, 33, 35).

Round 2

Reviewer 2 Report

the authors responded appropriately to the reviewer comments.

Author Response

Thank you very much for your comments and suggestions.

Reviewer 3 Report

The authors revised according to the suggestions made and answered to all the questions.

Author Response

(The authors gave the same response as above.)
